# Naturally Occurring *Fusarium* Species and Mycotoxins in Oat Grains from Manitoba, Canada

**DOI:** 10.3390/toxins13090670

**Published:** 2021-09-18

**Authors:** M. Nazrul Islam, Mourita Tabassum, Mitali Banik, Fouad Daayf, W. G. Dilantha Fernando, Linda J. Harris, Srinivas Sura, Xiben Wang

**Affiliations:** 1Agriculture and Agri-Food Canada (AAFC), Morden Research and Development Centre, 101 Route 100, Morden, MB R6M 1Y5, Canada; naz.islam@agr.gc.ca (M.N.I.); mitali.banik@agr.gc.ca (M.B.); srinivas.sura@agr.gc.ca (S.S.); 2Department of Plant Science, University of Manitoba, 66 Dafoe Road, Winnipeg, MB R3T 2N2, Canada; tabassu3@myumanitoba.ca (M.T.); fouad.daayf@umanitoba.ca (F.D.); dilantha.fernando@umanitoba.ca (W.G.D.F.); 3Agriculture and Agri-Food Canada (AAFC), Ottawa Research and Development Centre, 960 Carling Avenue, Ottawa, ON K1A 0C6, Canada; linda.harris@agr.gc.ca

**Keywords:** *Fusarium* head blight, oats, mycotoxins, chemotypes, phylogenetic analysis

## Abstract

*Fusarium* head blight (FHB) can lead to dramatic yield losses and mycotoxin contamination in small grain cereals in Canada. To assess the extent and severity of FHB in oat, samples collected from 168 commercial oat fields in the province of Manitoba, Canada, during 2016–2018 were analyzed for the occurrence of *Fusarium* head blight and associated mycotoxins. Through morphological and molecular analysis, *F. poae* was found to be the predominant *Fusarium* species affecting oat, followed by *F. graminearum*, *F. sporotrichioides*, *F. avenaceum*, and *F. culmorum*. Deoxynivalenol (DON) and nivalenol (NIV), type B trichothecenes, were the two most abundant *Fusarium* mycotoxins detected in oat. Beauvericin (BEA) was also frequently detected, though at lower concentrations. Close clustering of *F. poae* and NIV/BEA, *F. graminearum* and DON, and *F. sporotrichioides* and HT2/T2 (type A trichothecenes) was detected in the principal component analysis. Sampling location and crop rotation significantly impacted the concentrations of *Fusarium* mycotoxins in oat. A phylogenetic analysis of 95 *F. poae* strains from Manitoba was conducted using the concatenated nucleotide sequences of *Tef-1α*, *Tri1*, and *Tri8* genes. The results indicated that all *F. poae* strains belong to a monophyletic lineage. Four subgroups of *F. poae* strains were identified; however, no correlations were observed between the grouping of *F. poae* strains and sample locations/crop rotations.

## 1. Introduction

Canada is one of the world’s leading countries in the production and export of high-quality oat (*Avena sativa L*.), accounting for 15% of total global production and approximately 60% of global exports [1]. Oat is grown for both feed and food in Canada. In recent years, the use of oat and processed oat products for human consumption has increased by approximately 5% on an annual basis due to its unique nutritional benefits, such as high β-glucan content, high protein content, and healthy lipid profile [1,2].

Fusarium head blight (FHB) is a severe production problem for wheat (*Triticum aestivum* L.) and barley (*Hordeum vulgare* L.) producers in North America. In comparison, oat has been considered as a less susceptible host to FHB; this is mainly attributed to the lack of visual symptoms on infected panicles and long pedicels between spikelets that prevent the spread of fungal mycelia throughout the panicle [3,4]. However, oat grains can accumulate considerable amounts of *Fusarium* mycotoxins. FHB on oats in western Canada was first identified during the FHB epidemic in 1993. Since then, the sporadic and localized presence of FHB on oat has been reported in both western and eastern provinces of Canada [4,5,6,7]. A gradual increase in the incidence of FHB on oat in Canada has been documented in recent years; however, a severe FHB outbreak in oat has not yet been recorded [8].

FHB is caused by several species of fungi in the genus *Fusarium*. *F. graminearum* (Schwabe) and *F. poae* (Peck) Wollenw. are the principal *Fusarium* species causing FHB on oat in the prairie and eastern provinces of Canada. Several other species, including *F. sporotrichioides* Sherb., *F. avenaceum* (Fr.) Sacc., *F. culmorum* (W.G. Smith) Sacc., and *F. equiseti* (Corda) Sacc., are isolated from oat grains but at much lower frequencies [5,8]. *F. graminearum* is recognized as the most virulent species and the principal causal agent of FHB worldwide, including in Canada [9,10]. *F. poae* is generally regarded as a weak pathogen to cereals; however, it is the most frequently isolated *Fusarium* pathogen in several cereal disease surveys conducted across South American and European countries [11,12]. Additionally, *F. sporotrichioides* was also recognized as one of the causative agents of FHB in temperate regions of Europe [13]. 

FHB-infected cereal grains are often contaminated with mycotoxins that are toxic to animals and humans. *Fusarium* species can produce multiple mycotoxins. *F. graminearum* is a potent producer of deoxynivalenol (DON), the mycotoxin most commonly detected in cereal grains in Canada. DON is known to induce toxicity by inhibiting protein synthesis in both humans and animals [10,13]. *F. poae* can produce both trichothecene mycotoxins, including diacetoxyscirpenol (DAS), monoacetoxyscirpenol (MAS), scirpentriol (STO), nivalenol (NIV), fusarenone-X (FX), and non-trichothecene mycotoxins, such as beauvericin (BEA) [12,14,15]. NIV is the most documented mycotoxin produced by *F. poae* [14,16]. It is lymphotoxic to human B and T cells and can also induce apoptosis in human promyelocytic leukemia cells [17]. In Europe, T-2 and HT-2 are trichothecenes commonly detected in oat. They are potent cytotoxic and immunosuppressive agents that can cause acute intoxication and chronic diseases in both humans and animals [13]. An outbreak of alimentary toxic aleukia due to T-2/HT-2 contamination killed thousands of people in Russia in the 1940s [18]. *F. langsethiae* and *F. sporotrichioides* are potent producers of T-2 and HT-2 [19]. In Scandinavian countries, *F. langsethiae* is the main contributor to the accumulation of these mycotoxins in oat grains [20]. However, to the best of the authors’ knowledge, this pathogen has not been identified in Canada. 

The multi-species nature of FHB on oat significantly impacts current management practices for this disease. *F. avenaceum*, *F. langsethiae*, and *F. poae* are more tolerant to ergosterol-biosynthesis-inhibiting azole fungicides than *F. graminearum* [21]. Evidence suggests the efficacy of fungicides and biological control treatments against *F. graminearum* is reduced when other *Fusarium* pathogens co-occur at infection sites [22]. Furthermore, the application of certain fungicides may lead to an increased share of other *Fusarium* species in the FHB population at the expense of *F. graminearum* [23]. A detailed understanding of the spectrum of *Fusarium* species associated with FHB in oat and the naturally occurring mycotoxins associated with these pathogens is urgently needed to prevent crop contamination. 

To investigate the prevalence of FHB on oat and the disease complexity in terms of species and mycotoxin profiles, we analyzed oat samples collected from commercial fields from 2016 to 2018. Our study aimed to investigate: (i) *Fusarium* species spectrum and main *Fusarium* chemotypes associated with FHB in oat; (ii) the profile of naturally occurring *Fusarium* mycotoxins in oat grains and main contributors of these mycotoxins; (iii) the phylogenetic relationship of a collection of *F. poae* strains from Manitoba; (iv) the impact of sample location and crop rotation on mycotoxin levels in oat samples.

## 2. Results

Due to the lack of typical FHB symptoms on infected oat panicles, *Fusarium* infection in oat was investigated using the plating method and PCR-based techniques using *Fusarium* species-specific primers. Five *Fusarium* species were isolated from oat grains collected in Manitoba in 2016–2018 (Table 1). *F. poae* was the predominant species, occurring in 68.7% of fields and 58.7% of kernels. Infection caused by *F. graminearum* was also common, detected in 33.7% of fields and 16.6% of kernels. In comparison, the infection caused by *F. sporotrichioides* was less frequent, found in 10.3% of fields and 14% of kernels. *F. avenaceum* and *F. culmorum* were only sporadically detected (Table 1).

A PCR-based method was used to verify the spectrum of *Fusarium* species in oat samples. *F. poae* was the most frequently detected *Fusarium* species (59%), followed by *F. graminearum* (25%) and *F.*
*sporotrichioides* (12%). These three *Fusarium* species accounted for 97% of all detected *Fusarium* species (Figure 1A). Both 3-ADON and 15-ADON chemotypes of *F. graminearum* were detected (Figure 1B)*. F. graminearum* 3-ADON strains were detected in 49% of fields, whereas 15-ADON strains were detected in 45% of fields. We did not detect any *F. graminearum* strains with the NIV chemotype. However, *F. poae* strains with the NIV chemotype were detected in 91% of fields (Figure 1B). 

Quantitative PCR (qPCR) was used to quantify the abundance of *Fusarium* DNA in oat grains (Table 2). Due to low Ct values (<37) in the majority of samples, data for *F. avenaceum* and *F. culmorum* were not included (Table 2). The concentration of *F. poae* DNA in oat samples was the highest (Table 2). For the 2016 and 2017 samples, the concentration of *F. poae* DNA was two times higher than that of *F. graminearum*. This result agrees with the findings of the morphological analyses. The mean concentration of *F. graminearum* DNA was lower than that of *F. poae* but significantly higher than for *F.*
*sporotrichioides*. 

Naturally occurring mycotoxins in oat grains were analyzed using an ultra-high-performance liquid chromatography-high-resolution mass spectrometry (UHPLC-HRMS). Deoxynivalenol, NIV, and BEA were the three most common mycotoxins found in oat grains from Manitoba (Table 3). The maximum concentrations of DON in oat samples were 4143, 1881, and 632 ppb in 2016, 2017, and 2018, respectively. The mean concentration of DON was the highest in 2016 (604 ppb) and the lowest in 2018 (253 ppb). NIV was detected in 75 to 100% of samples, with mean concentrations of 252, 234, and 145 ppb for 2016, 2017, and 2018, respectively. BEA was detected in 70 to 90% of samples. The mean concentrations of BEA were lower than those of DON and NIV. Contamination of oat grains with T-2, HT-2, Moniliform (MON), DAS, and Enniatins (ENNs) (A, A1, B, and B1) only occurred at very low levels (Table 3). 

Principal component analysis (PCA) was performed to identify the primary contributors of different *Fusarium* mycotoxins in oats. The first two dimensions accounted for 53% of the variability using the three most common *Fusarium* species (*Fp*, *Fg*, and *Fs*) found in the grain samples. On the *x*-axis, component 1 describes 31% of the variability; on the *y*-axis, component 2 represents an additional 22% of the original variability (Figure 2). A strong correlation was found between *Fg* DNA and DON as they clustered in the same quadrant. Similarly, *Fp* DNA, NIV, and BEA showed a close association. *Fs* and T-2/HT-2 showed the maximum correlation within 3-year sample analyses (Figure 2). MON, DAS, and ENNs were excluded in the correlation analysis due to the low detection rate in the majority of oat samples.

We investigated the impact of sampling locations and crop rotation on mycotoxin levels in oat grains. The ANOVA showed sampling location impacted the mean concentrations of DON, NIV, BEA, T-2, and HT-2 (Table 4). The concentration of DON was the highest in samples from Central Manitoba (CMB) and Eastern Manitoba (EMB), and the concentration of NIV was higher in samples from CMB and SWMB (Southwest Manitoba). The average concentration of BEA in samples from CMB and INMB (Interlake Manitoba), T-2 in samples from CMB, and HT-2 in CMB and SWMB were higher compared to other crop districts. Crop rotation also had a significant impact on the mean concentrations of DON, BEA, and T-2 (Table 4). The concentrations of DON, NIV, BEA, and T-2 were the highest in grain samples from fields undergoing cereal (wheat/barley)-oat rotation (Figure 3). However, these toxins were measured at significantly lower levels when canola *(Brassica napus* L.) or flax *(Linum usitatissmum* L.) was grown instead of cereal (wheat/barley) as a preceding crop.

The concatenated nucleotide sequences of genes encoding a Translation elongation factor 1-alpha *(Tef-1α)*, a P450 oxygenase (*Tri1*), and a trichothecene 3-O-esterase *(Tri8)* were used to infer phylogenetic relationships among 95 *F. poae* strains collected in Manitoba using the Maximum Likelihood (ML) method. The results of the phylogenetic analysis indicated that all *F. poae* strains belonged to a monophyletic lineage (Figure 4). With a strong bootstrap value support (>75%), the consensus tree from ML analysis suggested four groups of *F. poae* strains. No clear correlation was evident between the grouping of these *F. poae* strains and sampling locations or previous hosts.

## 3. Discussion

This study investigated the *Fusarium* species complex associated with FHB on oat in Manitoba. The results show FHB is common on oat in Manitoba, with *F. poae* and *F. graminearum* being the two most commonly isolated species. In Manitoba, oat is grown in the same areas as wheat and barley, for which *F. graminearum* is the predominant *Fusarium* species found in Canada [8,10]; therefore, it is not surprising that *F. graminearum* is commonly isolated from oat grains. *F. graminearum* strains with both 3-ADON and 15-ADON chemotypes were detected in oat samples, with 3-ADON strains found at a higher frequency. This pattern is comparable to the chemotype structures of *F. graminearum* previously reported in wheat. Prior to 1994, *F. graminearum* strains with the 15-ADON chemotype were the principal pathogen causing FHB in wheat in North America; however, a shift to *F. graminearum* strains with the 3-ADON chemotype was documented between 1998 and 2004 [24]. Several studies show a higher percentage of *F. graminearum* strains with 3-ADON vs. 15-ADON chemotypes in wheat from Western Canada [24,25,26]. Ward et al. [27] hypothesize that Eastern Canada was the original point of entry for *F. graminearum* 3-ADON populations in North America, followed by a spread into Western Canada and the upper Midwest of the United States. The drivers of the shifts in *F. graminearum* chemotype are not known but may be influenced by environmental conditions, host distribution, and various agricultural practices [28]. 

On wheat, *F. poae* is often considered a weakly pathogenic species. It is long thought that *F. poae* may infect and grow superficially but does not proliferate beyond infection sites [29,30]. Our study shows a higher concentration of *F. poae* DNA in oat grains than *F. graminearum* DNA. This indicates *F. poae* is well established on oat under field conditions. To date, information regarding the pathogenicity of *F. poae* on oat and the mechanism by which it interacts with other *Fusarium* species during infection has been limited. *F. poae* requires dry and warm conditions of around 25 °C for optimum growth and infection. In contrast, *F. graminearum* infection is often associated with prolonged wet and warm conditions during anthesis [31]. Environment parameters may differentially impact the success of these *Fusarium* pathogens on oat. Another possibility is that *F. poae* has a competitive advantage when it co-occurs with other *Fusarium* species during the infection. Ameye et al. [32] demonstrated that the volatile organic compounds emitted by perennial ryegrass (*Lolium perenne* L.) infected with *F. poae* could protect the plants against *F. graminearum* by priming for jasmonic acid (JA)-dependent defence. More recently, Tan et al. [33] showed that the expression of genes encoding isochorismate synthases and lipoxygenases is upregulated during the asymptomatic infection of F. poae on wheat; they hypothesize *F. poae* could hamper a subsequent *F. graminearum* infection by inducing a host defence response. Additionally, there is evidence suggesting that *F. poae* is more resistant to ergosterol biosynthesis-inhibiting azole fungicides than *F. graminearum* [34]. The application of azole fungicides leads to an increased share of *F. poae* in the FHB population at the expense of *F. graminearum* in the field [23]. Thus, the selective pressure exerted by the extensive use of certain fungicides may also contribute to the predominance of *F. poae* on oat. 

Deoxynivalenol was one of the most prevalent mycotoxins in oat from the present study, mainly associated with the infection caused by *F. graminearum*. Similarly, Gräfenhan et al. [35] demonstrated that DON is the most frequently detected mycotoxin in oat from Western Canada. In a more recent survey of FHB on oat in Eastern and Western Canada, DON was commonly detected in oat grains from these regions [36]. In Canada, the maximum limits for DON in uncleaned soft wheat are 2000 and 1000 ppb for staple foods and infant foods, respectively (https://www.grainscanada.gc.ca/en/about-us/consultations/2019/falling-number-don.html, accessed 10 July 2021). The European Commission has set 1750 ppb as the maximum limit for DON in unprocessed oat [37]. Oat samples from this study from both 2016 and 2017 exceed these DON concentration limits; however, sampling focused on FHB-infected spikes. Oats grown in Canada are mainly used for animal feed and human consumption, so FHB monitoring is also important in oat to reduce the threat of *F. graminearum* and DON to oat industries in Canada. 

Nivalenol was found to be a common contaminant in oat from all sampling districts. This result is in agreement with studies of Tittlemier et al. [36] and Campbell et al. [38], in which NIV was detected second only to levels of DON in oat grains collected from Western Canada. DON and NIV differ in their chemical structure; NIV has a hydroxyl group instead of hydrogen at the C4 position, increasing its toxicity to animals and humans compared to DON [39,40]. Additionally, several studies indicate that the co-occurrence of DON and NIV will synergistically affect their cytotoxicity [41,42]. The present study did not find any *F. graminearum* strains with the NIV chemotype in oat samples, but *F. poae* strains with the NIV chemotype were commonly detected. The concentrations of NIV and *F. poae* DNA in oat grains are highly correlated. Therefore, we conclude that *F. poae* is the main contributor of NIV in these oat samples from Manitoba. Similarly, Yli-Mattila et al. [43] and Schöneberg et al. [2] show high correlations between *F. poae* DNA and NIV levels in cereal grains. *F. poae* strains producing NIV in oat grains are concerning because testing for NIV is not routinely conducted at mills or elevators in Canada at present. Our study shows the importance of testing for NIV in naturally infected oat grains and a more extensive study on the importance of *F. poae* in *Fusarium* species complex affecting oats. 

In European countries, HT-2/T-2 are the predominant mycotoxins found in oat grains, and *F. langsethiae* is the main contributor [2,44]. In this study, mean levels of T-2/HT-2 were much lower than those of DON and NIV. To date, *F. langsethiae* has not been identified in Canada to the best of the authors’ knowledge. We observed a strong correlation between the concentration of *F. sporotrichioides* DNA and T-2/HT-2, indicating *F. sporotrichioides* is likely the primary producer of T-2/HT-2 in Manitoba oats. The Canadian Food Inspection Agency has recommended limiting the intake of HT-2 to 0.1 ppm for cattle and poultry; the recommended tolerance level for T-2 is <1 ppm (https://inspection.canada.ca, accessed 10 July 2021). A few samples from 2016 had T-2 and/or HT-2 levels above 1000 μg/kg. With FHB becoming a severe issue for oat in Canada in recent years, closely monitoring the levels of T-2 and HT-2 in naturally infected oat grains will be very important. 

The ANOVA revealed a significant impact of sampling location on mycotoxin levels in oat. Several studies report significant variation in DON concentrations among grain samples collected from different regions of Canada. Environmental factors during anthesis, such as temperature and precipitation, can significantly impact the severity of FHB and the level of DON [6,38,45]. It is likely that higher mycotoxin concentrations in oat samples from CMB, SWMB, and EMB are due to the localized climate in these regions, which were more conducive for the infection of *Fusarium* pathogen, such as the extended period of rainfall during the stage of anthesis. Other factors could also impact mycotoxin levels in oat, such as seeding date, resistance in oat cultivars, and timing of fungicide applications. 

Crop rotation also impacts mycotoxin levels in oat grains. In our study, the cereals-oat rotation resulted in higher mycotoxin levels in oat grains than that with canola–oat or flax–oat rotations. Similarly, Pageau et al. [46] showed that the highest DON content in barley occurs when cereals rather than a dry pea are seeded in crop rotation. The preceding crop likely affects FHB epidemics by acting as a suitable host plant for *Fusarium* pathogens, i.e., increasing the amount of inoculum and producing large amounts of crop debris suitable for the saprophytic survival of these pathogens [47]. Our result indicates that crop rotation will have a significant impact on the management of FHB. Rotating away from cereals to non-host crops, including canola and pulses, could reduce the severity of FHB/mycotoxin contamination in oat by reducing the buildup of FHB infested crop residues from the previous growing season. We assessed the genetic variation of a set of *F. poae* strains collected in Manitoba using the concatenated sequence of *Tef-1α*, *Tri1*, and *Tri8* genes. The phylogenetic analysis indicates all of these *F. poae* strains are part of a monophyletic lineage. This result is in line with previous reports for *F. poae* [48]. No correlation between sample location and sub-grouping of *F. poae* strains was observed (Appendix A). Similarly, Dinolfo et al. [49] and Vanheule et al. [50] demonstrate that the clustering of *F. poae* strains based on ISSR and AFLP markers had no clear correlation with geographic origins. On the other hand, Vogelgsang et al. [51] characterize two groups of *F. poae* strains from distinct geographic areas within Switzerland using microsatellite markers. *F. poae* is now known to possess four core chromosomes and several supernumerary chromosomes [52]. The discrepancies among the results of the above studies may be due to the different types of genetic markers used, which target different regions of the *F. poae* genome. Witte et al. [53] show an Eastern Canadian *F. poae* strain (DAOMC 252244, *Fp*157) contains four core chromosomes and seven additional contigs associated with accessory chromosomes that harbour a functional biosynthetic apicidin synthetase (*APS1*) gene cluster, a hemorrhagic factor for toxin-producing fungi [54]. Such toxin potentiality of *F. poae* calls for further exploration of *APS*-producing populations in Western Canadian oat production systems. 

In *Fusarium* species, the production of trichothecenes is a result of the expression of the *Tri* 12-gene cluster and the unlinked *Tri1* and *Tri101* genes. In *F. graminearum*, variations in *Tri1* and *Tri8* are associated with the production of different trichothecene chemotypes [55,56]. Vanheule et al. [50] report that the *Tri1* locus in *F. poae* is highly variable with mostly synonymous mutations, with two main *Tri1* types identified in a collection of European *F. poae* strains. In this study, we observed reasonably high intra-species variability of *Tri1* and *Tri8* in *F. poae* strains from Manitoba. Whether the variability of these two genes will affect the toxigenic potential of *F. poae* is currently unclear and is an interesting question that merits further investigation. The preliminary phylogenetic analysis conducted in this study serves as an essential addition to existing knowledge on the diversity of the *F. poae* population in Manitoba. The results will facilitate the selection of *F. poae* strains for future genomic studies to better understand this emerging pathogen’s diversity, genomics, and toxic potential. 

## 4. Materials and Methods

### 4.1. Oat Sample Collections

A total of 168 oat fields (43, 60, and 65 in 2016, 2017, and 2018, respectively) in Manitoba were surveyed for FHB between July 18 and August 5 of each year. The field locations were divided into five crop districts: (1) Central Manitoba (CMB), (2) Southwest Manitoba (SWMB), (3) Eastern Manitoba (EMB), (4) Northwest Manitoba (NWMB), and (5) Interlake Manitoba (INMB) (Appendix A). The oat fields were surveyed between the late milk growth stage (BBCH 77) and full maturity (BBCH 89). Fields were accessed approximately every 20–25 km along the survey routes, depending on crop availability and accessibility. On average, 40–60 FHB infected spikes/panicles were randomly collected from each oat field and stored in paper envelopes. For each sampling location, information on the preceding crop was also collected.

### 4.2. Isolation and Identification of Fusarium Pathogens

For each field, 50 oat seeds were randomly selected and used to isolate *Fusarium* pathogens using Potato Dextrose Agar (PDA, Thermo Fisher, Mississauga, ON, Canada) media. Before plating, oat seeds were surface-sterilized for 90 s in 0.3% sodium hypochlorite, rinsed twice with distilled water, and then dried for 5 min at the room temperature. The PDA plates were incubated for 5 days at room temperature (20–25 °C) in the dark. A total of 8400 oat grains over 3 years were subjected to the identification of *Fusarium* species. The identification of *Fusarium* pathogens was performed using compound and dissecting microscopy based on conidia morphology [57]. 

### 4.3. Extraction and Quantification of Fusarium Genomic DNA in Oat Grains

A subsample of 20 g oat grains per field was ground with an oat grinder (Retsch ZM 200, Scientific Inc. Newtown, PA, USA). One-gram grain flour was used to extract DNA using the QIAGEN DNeasy Mini Kit (QIAGEN Mississauga, ON, Canada) following the manufactures’ procedure.

Quantitative PCR was used to determine the abundance of *Fusarium* genomic DNA in oat using a CFX96™ Real-Time PCR Detector System (BioRad, Mississauga, ON, Canada). All standards and the negative control (double-distilled water) were run in triplicate. Primers, based on the elongation factor 1α (*Tef-1α*) gene for three *Fusarium* species (*F. graminearum*, *F. poae*, and *F. sporotrichioides*), were used to quantify *Fusarium* genomic DNA in oat grains (Appendix A). The PCR was carried out in a total volume of 20 μL, consisting of 10 μL of SsoFast EvaGreen^®^ PCR Master Mix (BioRad), 1 μL of each primer, 6 μL of double-distilled water, and 2 μL of template DNA with a 37-cycle threshold (Ct) detection limit. PCR conditions were as follows: initial preheating at 98 °C for 2 min; 40 cycles of 95 °C for 15 s and 62 °C for 1 min; and dissociation curve analysis at 60 to 95 °C. Six technical replicates were performed for each sample. 

### 4.4. Identification of Fusarium Chemotypes 

Multiplex PCR was performed to determine 3-ADON, 15-ADON, and NIV chemotypes with specific primers (Appendix A). PCR assays were completed using the following cycle parameters: initial denaturing step at 95 °C for 5 min; 30 cycles at 95 °C for 30 s, 50 °C for 30 s, and 72 °C for 2 min; and a final extension step at 72 °C for 3 min. PCR amplicons were separated on 2% agarose gels in the 1 × TAE buffer and stained with GelRed (Biotium, Mississauga, ON, Canada). Gel images were scanned into Gel Doc™ EZ Imager (BioRad). Percentages of oat fields infected with *Fusarium* pathogens with 3-ADON, 15-ADON, and NIV chemotypes were recorded. 

### 4.5. Detection and Quantification of Mycotoxins

Oat grain samples were analyzed for 11 *Fusarium* mycotoxins including DON, DAS, NIV, BEA, HT-2, T-2, moniliformin (MON), enniatin A (ENN A), enniatin A1 (ENN A1), enniatin B (ENN B), and enniatin B1 (ENN B1) at the Mycotoxin Lab (Morden Research and Development Centre, AAFC, MB, Canada). Ground oat samples (1 g) were used to extract mycotoxins using a mixture of solvent (10 mL, acetonitrile (75%): methanol (10%): water (15%) in a 10 mL flat-bottomed tube. Sample–solvent mixture was mixed thoroughly by inversion, followed by sonication (30 min). The tubes were then loaded onto a rotatory shaker and extracted for 90 min at 40 rpm. The samples were centrifuged (4000× *g* for 30 min) to separate the extract from the ground material, and the supernatant was filtered through a 0.2 µm nylon syringe filter (ThermoFisher, Mississauga, ON, Canada) into a clean 10 mL flat-bottomed tube. The filtered extracts were dried under a gentle stream of nitrogen gas in a nitrogen evaporator (RapidVap Labconco, Kansas city, MO, USA) at 100% speed for 90 min at 70 °C. Each dried-sample extract was re-suspended in 1 mL of 50:50 water + 0.1% formic acid + 5 mM ammonium formate): methanol + 0.1% formic acid + 5 mM ammonium formate, vortexed, and transferred into amber liquid chromatography (LC) vial before analyzing using a high-resolution mass spectrometer (Orbitrap ID-X Tribrid Mass Spectrometer, Thermo Fisher Scientific Inc., Mississauga, ON, Canada) coupled with ultra-high performance liquid chromatography (UHPLC-HRMS, Vanquish, Thermo Fisher Scientific, Mississauga, ON, Canada). Separation of various mycotoxins was performed using a reverse-phase C18 core–shell silica column (particle size 1.7 µm, 100 × 2.1 mm, Kinetex, Phenomenex, CA, USA) held at 35 °C. Gradient elution was achieved with 100% water (mobile phase A) and 100% methanol (mobile phase B) with both phases containing 0.1% formic acid and 5 mM ammonium formate and at a flow rate of 0.2 mL min^−1^ in 20 min runs. Retention times are identified in Appendix A. Heated-electrospray (H-ESI) was used to achieve a steady state of electrospray from the chromatographic separation column. The HRMS was operated in positive mode at 120,000 orbitrap resolution for precursor ion mass-to-charge (*m/z*) with mass tolerance of ±5 ppm while fragment ion mass spectra were acquired at 30,000 resolution. Precursor and fragment ion *m/z* values are noted in Appendix A. Fragmentation was achieved in the higher energy collision-induced dissociation (HCD) with stepped collision energies (15, 20, and 25%). The parent mass was used for analyte quantification, and fragment ions were used for analyte confirmation.

Spiked reference material was extracted with each batch of samples and was extracted simultaneously. The recoveries for various mycotoxin standards (Sigma-Aldrich, Oakville, ON, Canada) are noted in Appendix A. The limit of quantification (LOQ) and limit of detection (LOD) for various mycotoxins were 50 and 10 ppb, respectively.

### 4.6. Sequencing of F. poae Tef-1α, Tri1, and Tri8 Genes

DNA was extracted from mycelium of *F. poae* using the QIAGEN DNeasy Mini Kit. Sequences of *Tef-1*α, *Tri1*, and *Tri8* were amplified directly from *F. poae* genomic DNA using the primer sets listed in Appendix A. Primers were designed based on the consensus of previously published *F. poae* genome sequences [53]. PCR reactions were performed with a C1000 Touch™ Thermal Cycler system (BioRad) in a volume of 25 μL containing 2.5 μL of 10 × PCR buffer, 15.88 μL of water, 2 μL of dNTP mix (2.5 mM), 1 μL of each primer (10 pmol), 0.125 μL of Taq polymerase (5 U/μL), and 2.5 μL of template DNA (100 ng/μL). Amplification was carried out using an annealing temperature of 56 °C for 30 s. The PCR assay was completed using the following cycle parameters: initial denaturing step at 94 °C for 15 min; 30 cycles at 95 °C for 30 s, 50 °C for 30 s, and 72 °C for 90 s; and a final extension step at 72 °C for 3 min. The PCR products were separated electrophoretically on 1.5% agarose gels in 1 × TAE buffer and stained with gel Red (Biotium). The amplified bands were cut under UV light (UVITEC, Rugby, UK), and purification of the gene (PCR amplicons) was performed using Gel Extraction Kits (Qiagen) following the manufacturer’s protocols. The concentration of the products was checked using a NanoDrop 2000 spectrophotometer (ThermoFisher, Wilmington, NC, USA). The cleaned PCR amplicons were sequenced using Sanger sequencing protocols (ABI 3730xl DNA sequencers, National Research Council, Saskatoon, SK, Canada). 

### 4.7. Phylogenetic Analysis of F. poae Strains Based on Tri1, Tri8, and Tef-1α Genes

The raw Sanger sequences were processed using MEGAX to clean ambiguous nucleotides (average quality >50). Excessively long and short homopolymer, low-quality, and chimeric sequences were removed from the dataset. The alignments of *Tef-1α*, *Tri1*, and *Tri8* were created using MUSCLE [58] with default settings. TrimAI [59] was used to trim the alignments and remove positions with gaps in 10% or more of the sequences. The concatenated sequence of *Tef-1α*, *Tri1*, and *Tri8* of *F. graminearum* PH-1 (assembly ASM24013v3) was also created using MEGAX and included as an outgroup. A partitioned file stating the positions of individual genes in the concatenated alignment was created. IQTree (version 1.6.12) was used to find the best substitution model for each partition (-m MFP) and infer a consensus tree using maximum likelihood and ultrafast bootstrapping (*n* = 1000).

### 4.8. Statistical Analysis

The homogeneity of data (*n* = 168 oat fields/samples) was verified using the Kolmogorov–Smirnov test of normality, which measured the divergence of the field sample distribution. The skewness and kurtosis were determined and plotted for each year of survey data to determine whether the data distribution was normal. The data were transformed (log, square root, arcsine), and outliers were removed (extreme low and high values) when necessary. One-way analysis of variance (ANOVA) (Version 9.4, SAS Institute, Cary, NC, USA) was used to determine the significance of survey years, geographical field locations/crop districts, and preceding crops (crop rotation) on the concentration of five mycotoxins (DON, NIV, BEA, HT-2, T-2). The Least Significant Difference (LSD) test was used to compare the means of different treatment groups. Principal component analysis (PCA) was performed to analyze the correlation between DNA concentrations of three *Fusarium* species (*F. poae*, *F. graminearum*, *F. sporotrichioides*) and five main *Fusarium* mycotoxins (DON, NIV, BEA, HT-2, T-2) using SAS software statistics. 

## Figures and Tables

**Figure 1 toxins-13-00670-f001:**
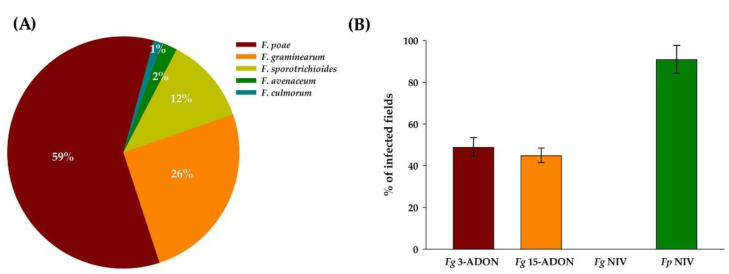
PCR-based characterization of *Fusarium* species complex associated with *Fusarium* head blight on oat in Manitoba (2016–2018). (**A**): Frequencies of *Fusarium* DNA detected in commercial oat fields in Manitoba; (**B**): distribution of *Fusarium* chemotypes in Manitoba oat fields. Number of samples (*n*) = 168.

**Figure 2 toxins-13-00670-f002:**
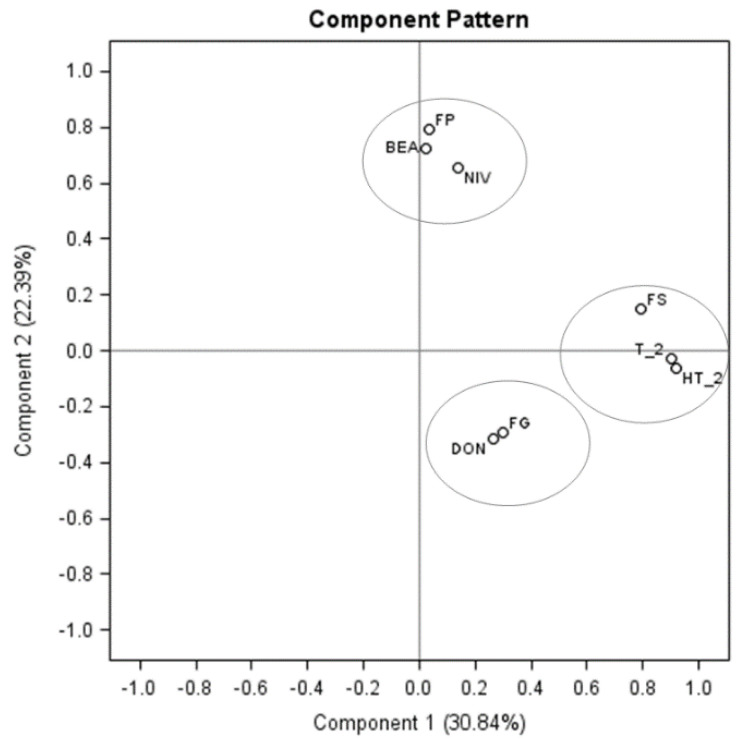
Principal component analysis of the relationship between the abundance of *Fusarium* DNA (FP: *F. poae*, FG: *F. graminearum,* and FS: *F. sporotrichioides*) and mycotoxin concentrations (NIV, BEA, DON, T-2, and HT-2) in oat grains. The percentages of variance explained by axes/components 1 and 2 are shown in parentheses. Each point represents the mean of 3-year samples (*n* = 168).

**Figure 3 toxins-13-00670-f003:**
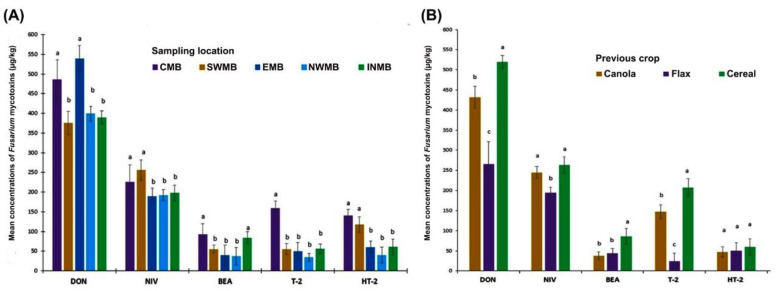
The effect of sampling location and crop rotation on the three-year mean concentration of *Fusarium* mycotoxins in oat samples from Manitoba. Bars (mean ± standard error) followed by the same letter are not significantly different at *p* = 0.05; (**A**) The samples locations were divided into fixed crops districts including CMB (Central Manitoba), SWMB (Southwest Manitoba), EMB (Eastern Manitoba), NVMB (Northwest Manitoba), and INMB (Interlake Manitoba). (**B**) The effect of previous crop on *Fusarium* mycotoxin content in Manitoba oat samples, collected in 2016, 2017, and 2018.

**Figure 4 toxins-13-00670-f004:**
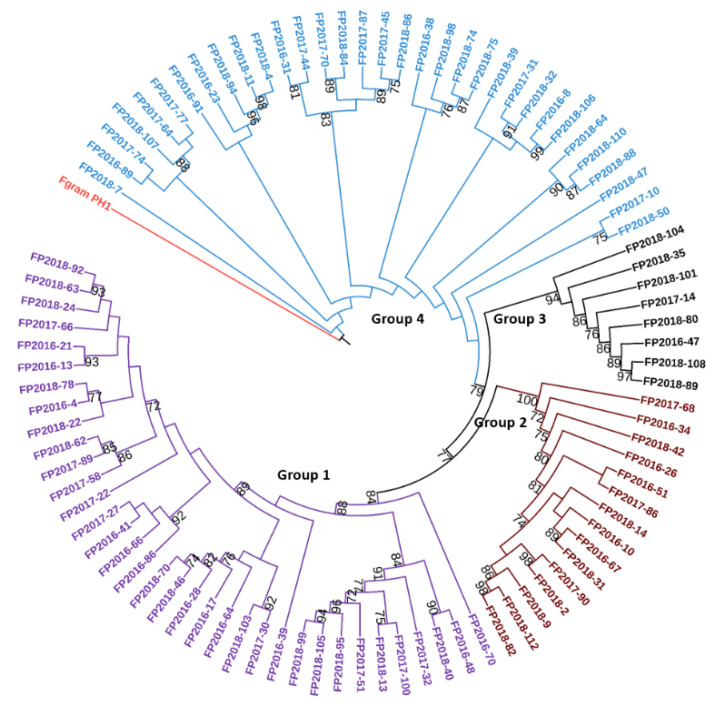
Maximum likelihood phylogenetic tree of 95 *F. poae* strains collected in Manitoba. The phylogenetic tree was inferred from the concatenated sequences of *Tef-1α*, *Tri1*, and *Tri8* genes. Relevant bootstrap values (expressed as a percentage of 1000 replicates) are shown at branch points. The concatenated sequence of *F. graminearum* strain PH1 obtained from GenBank was treated as the outgroup.

**Table 1 toxins-13-00670-t001:** *Fusarium* species complex in oat kernels collected from commercial fields in Manitoba (2016–2018).

*Fusarium* Species	% of Infected Fields				% of Infected Kernels			
	2016	2017	2018	Three-Year Mean	2016	2017	2018	Three-Year Mean
*F. poae*	72.0	65.0	69.0	68.7 ± 3.5	61.0	62.0	53.0	58.7 ± 4.9
*F. graminearum*	28.0	25.0	48.0	33.7 ± 9.6	22.3	14.8	12.8	16.6 ± 5.0
*F. sporotrichioides*	9.0	7.0	15.0	10.3 ± 4.2	8.0	22.0	12.0	14.0 ± 7.2
*F. avenaceum*	2.3	3.3	1.3	2.3 ± 11	0.4	0.5	0.2	0.4 ± 0.2
*F. culmorum*	2.3	1.7	0.0	1.3 ± 1.2	0.4	0.3	0.0	0.2 ± 0.2
Total no. of fields/kernels	43	60	65		2150	3000	3250	

**Table 2 toxins-13-00670-t002:** Quantitative PCR analysis of the abundance of *F. poae*, *F*. *graminearum*, and *F. sporotrichioides* DNA in oat grains from Manitoba (2016–2018).

Year	2016		2017		2018	
*n*-Samples/Fields	43		60		65	
Species	Range	Mean	Range	Mean	Range	Mean
*F. poae*	0.01–1.58 *	0.13 ± 0.05 a **	0.02–1.35	0.12 ± 0.07 a	0.00–1.37	0.08 ± 0.01 a
*F. graminearum*	0.01–1.93	0.05 ± 0.01 b	0.01–0.80	0.07 ± 0.01 a	0.01–0.31	0.07 ± 0.01 a
*F. sporotrichioides*	0.01 -0.38	0.02 ± 0.01 b	0.01–0.05	0.01 ± 0.01 b	0.00–0.15	0.01 ± 0.01 b

*** The abundance of *Fusarium* DNA is expressed as pg of *Fusarium* gDNA/ng of total gDNA. ** Mean ± SE refers to combining average concentrations of all samples. Means followed by the same letter within the same year are not significantly different at *p* = 0.05 based on Student’s *t*-test.

**Table 3 toxins-13-00670-t003:** Naturally occurring *Fusarium* mycotoxins in Manitoba oat grains (2016–2018).

Year	2016			2017			2018		
N-Samples/Fields	43			60			65		
	% of Fields Above LOD	Maximum (µg/kg)	Mean (µg/kg)	% of Fields Above LOD	Maximum (µg/kg)	Mean (µg/kg)	% of Fields Above LOD	Maximum (µg/kg)	Mean (µg/kg)
DON	85	4143	604 ± 87	41	1881	569 ± 104	40	632	253 ± 20
NIV	100	865	252 ± 27	78	795	234 ± 16	98	581	145 ± 12
T-2	77	1155	43 ± 18	81	973	53 ± 16	25	794	22 ± 12
HT-2	35	1100	60 ± 29	13	419	26 ± 10	17	654	23 ± 11
BEA	90	119	25 ± 4	80	159	41 ± 4	71	169	24 ± 5
MON	32	349	31 ± 11	20	533	30 ± 11	11	209	12 ± 5
DAS	15	25	8 ± 1.1	7	25	7 ± 0.6	5	25	6 ± 0.5
ENNs	43	1605	65 ± 6	39	93	16 ± 2	17	128	10 ± 1

Note: % of fields determined by mycotoxin infected fields/total number of fields. Mycotoxins presented here are at concentrations above the limit of quantitation (LOQ) and limit of detection (LOD) of 50 µg/kg (ppb) and 10 µg/kg (ppb), respectively. Mycotoxin concentrations below the respective LOQ or LOD were calculated as LOQ/2 or LOD/2, respectively, ± standard error. ENNs are the sum of ENN A, A1, B, and B1.

**Table 4 toxins-13-00670-t004:** ANOVA of the effect of sampling location/crop rotation on *Fusarium* mycotoxin levels in oat samples.

	^†^*p*-Value				
	*Fusarium* Mycotoxin Levels				
Source of variation	DON	NIV	BEA	T-2	HT-2
Sample locations	*	*	*	ns	*
Crop rotations	*	ns	*	*	ns

^†^*p*-values are marked as * (*p* < 0.05) to assess significant difference between *Fusarium* mycotoxin levels and sampling locations/crop rotations; ns indicates non-significant.

## Data Availability

Not applicable.

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
