# Peer review of "Naturally Occurring Fusarium Species and Mycotoxins in Oat Grains from Manitoba, Canada"

_toxins, 2021, doi:10.3390/toxins13090670_

Round 1
Reviewer 1 Report
Dear Authors,
The research topic is very interesting considering health consequences related to mycotoxins contamination in food. The scientific approach and the sample size are fully adequate to publication. Materials and Methods were generally well developed, but I suggest to briefly explaining some topic. The paper might be conform to journal format concerning layout and acronyms. Suggestions are reported in pdf-attached file. Some statistics flaws were in any graphs. Language level completely satisfy requested standard. Generally, I think that manuscript could be slightly improved in view of publication.
Kind regards.

Author Response
Reviewer 1:
The research topic is very interesting considering health consequences related to mycotoxins contamination in food. The scientific approach and the sample size are fully adequate to publication. Materials and Methods were generally well developed, but I suggest to briefly explaining some topic. The paper might be conformed to journal format concerning layout and acronyms. Suggestions are reported in pdf-attached file. Some statistics flaws were in any graphs. Language level completely satisfy requested standard. Generally, I think that manuscript could be slightly improved in view of publication.
Kind regards.
Our response:
Thanks for the encouragement and valuable suggestions. We have improved the manuscript based on the comment provided in the PDF file from reviewer 1. The correction that was made can be found in the track-changed version of the manuscript.
The major changes:
- The standard errors are included in the data from Fig. 1 (Pg 106 – 107).
- New figures are made for Figure 1. Both Fig. 1A and 1B are changed into coloured figures. The standard errors are included in Fig. 1B.
- The legend of Fig.3 is edited to better explain the graphs and related acronyms.

Reviewer 2 Report
The manuscript submitted for review entitled "Naturally occurring Fusarium species and mycotoxins in FHB-infected oat in Manitoba, Canada" presents interesting research results. The paper relates to:
- identification of Fusarium fungi by both microscopy and qPCR (by ef1α gene quantification),
- determination of the percentage of grains and ears infected by Fusarium
- determine the content of mycotoxins: DON, NIV, BEA, T2,HT2, MON, DAS and ENNs
- perform PCA analysis showing correlation between contents: DNA F.p and BEA/NIV; F.g and DON; F.s. and T2/HT2,
- by Multiplex PCR the chemotypes were determined: 3-ADON, 15-ADON and NIV
- statistical analysis was performed to show the effect of sampling location and crop rotation on mycotoxins content
- phylogenetic analysis of 95 F. poae isolates was performed.
The work is coherent, all analyses are very well done, and the conclusions are interesting although in a few cases not very revealing. It is necessary to explain the impact of location and what shifts influenced the contamination of individual mycotoxins. I think that work deserves to be published with minor revision.
- I believe the title is incorrect. FHB is a disease that does not cause infection. The infection is caused by Fusarium. I suggest the title: Naturally occurring Fusarium species and mycotoxins in FHB-symptomatic oat in Manitoba, Canada.
- I believe that symptoms of FHB were not investigated but only the occurrence of Fusarium fungi causing FHB on grain and ears. I found no description of the DIsease index of FHB in the methodology. Fungi from grain were isolated on PDA medium and PCR and qPCR analyses were performed. Please explain. If DI was tested very please describe it and attach photos of symptoms.
- Did the study isolate F. tricinctum, which is morphologically similar to F. pae and F. sporotichioides? PCR primers often cross F. tricinctum and F. avenaceum together.
- 4 Why didn't the authors use primers for ESYN1 gene detection when they isolated F. pae in abundance and showed significant levels of ENNs?
- L365- How were the infected FHB spikes selected?
- L366- On what basis were 8400 infected oat grains selected?
- What did the morphological identification of Fusarium consist of? Was it a mycological analysis as described below?
- Describe the DNA isolation.
- I think the conclusions about rotation and location are valuable, but need more explanation. What kind of rotation favored mycotoxin formation? What was the reason for the effect of location on mycotoxin formation? Climatic conditions? What kind of conditions? This needs to be clarified because it could be interesting.
Author Response
Reviewer 2:
Thanks for the valuable suggestions. We have revised the manuscripts based on the comments provided by reviewer 2.
Reviewer 2's comments:
- I believe the title is incorrect. FHB is a disease that does not cause infection. The infection is caused by Fusarium. I suggest the title: Naturally occurring Fusarium species and mycotoxins in FHB-symptomatic oat in Manitoba, Canada.
Our response:
It is an excellent point. We changed the title of the manuscript to "Naturally occurring Fusarium species and mycotoxins in oat samples from Manitoba, Canada." Since there is a lack of typical FHB symptoms on oat, the oat samples were randomly collected.
Reviewer 2's comment:
- I believe that symptoms of FHB were not investigated but only the occurrence of Fusarium fungi causing FHB on grain and ears. I found no description of the DIsease index of FHB in the methodology. Fungi from grain were isolated on a PDA medium, and PCR and qPCR analyses were performed. Please explain. If DI was tested very please describe it and attach photos of symptoms.
Our response:
Unlike FHB on barley and wheat, the typical FHB symptom on oat is rare in field conditions. It is challenging to rate FHB incidence and severity on oat using visual symptoms. Therefore, we didn't collect the FHB disease index data. We relied on the plating method using PDA medium and PCR-based detection method using Fusarium species primer sets to analyze the occurrence of FHB in oat samples included in our study.
Reviewer 2's comment:
- Did the study isolate F. tricinctum, which is morphologically similar to F. poae and F. sporotichioides? PCR primers often cross F. tricinctum and F. avenaceum together.w
Our response:
We didn't isolate any F. tricinctum strains from oat samples collected in this study. There has been no report of F. tricinctum in FHB on small grain cereals in Canada.
Reviewer 2's comment:
4 Why didn't the authors use primers for ESYN1 gene detection when they isolated F. poae in abundance and showed significant levels of ENNs?
Our response:
We tested the specificity of several sets of published Fusarium species-specific primer sets. Our data showed that the primers from the study of Demeke et al. (2005) gave us the best result for F. poae. It is highly specific to F. poae. We didn't obtain the data on ENNs until the late stage of our study. Therefore, we didn't use primers for the ESYN1 gene for the detection of F. poae. It is a fascinating suggestion, and we intend to investigate further in the future.
Reviewer 2's comment:
- L365- How were the infected FHB spikes selected?
- L366- On what basis were 8400 infected oat grains selected?
Our response:
Due to the lack of the typical FHB symptom on oat panicles, the samples were randomly collected. We analyzed 50 grains per field. A total of 168 fields was included in our analysis which added up to 8400 seeds. We have changed the material and method (Ln368 t0 382) to improve the clarity.
Reviewer 2's comment:
5 What did the morphological identification of Fusarium consist of? Was it a mycological analysis as described below?
Our response:
The morphological identification of Fusarium pathogens from the PDA plate was based on the morphology of Fusarium conidia. We have edited our material and method to provide a better clarity (Ln 378 to 381).
Reviewer 2's comment:
- Describe the DNA isolation.
Thank for the suggestion. We added the method of DNA isolation in ln 383-386 and ln 445.
Reviewer 2's comment:
- I think the conclusions about rotation and location are valuable, but need more explanation. What kind of rotation favored mycotoxin formation? What was the reason for the effect of location on mycotoxin formation? Climatic conditions? What kind of conditions? This needs to be clarified because it could be interesting.
Our response:
Thanks for the suggestion. We have explained the effect of rotation and location on mycotoxin level in Ln 318 to Ln320 and Ln 329-333.